# Assessing food consumed away from home in low-and middle-income countries by developing specific modules for household surveys: Experimental evidence from Vietnam and Burkina Faso

**Edwige Landais**[1]*, **Raphaël Pelloquin**[1], **Elodie Maître d'Hôtel**[1], **Mai Truong Tuyet**[2], **Nga Hoang Thu**[2], **Yen Bui Thi Thao**[2], **Ha Do Thi Phuong**[2], **Trang Tran Thi Thu**[2], **Jérôme Somé**[3], **Christophe Béné**[4], **Eric O. Verger**[1]

1 MoISA, Univ Montpellier, CIRAD, CIHEAM-IAMM, INRAE, Institut Agro, IRD, Montpellier, France, 2 National Institute of Nutrition, Nutrition-Network Department, Hanoi, Vietnam, 3 Département Biomédical et Santé Publique, Institut de Recherche en Sciences de la Santé, Centre National de la Recherche Scientifique et Technologie, Ouagadougou, Burkina Faso, 4 Food Environment and Consumer Behaviour Department, Alliance Bioversity International–CIAT, Cali, Colombia

* edwige.landais@ird.fr

**Data Availability Statement:** The present study involved human beings and the data collected are

## Abstract

In a world rapidly transitioning, food consumption away from home is rising, therefore representing an increasing share of individual's diet. Food consumed away from home negatively impacts diet, nutritional status and consequently has detrimental effects on health. In some contexts, where individual level dietary intake surveys are not regularly conducted, this behavior is not well documented leading to a gap of knowledge. The aim of the present study was to develop and validate in Burkina Faso and Vietnam specific modules that could be added to Household Consumption and Expenditure Surveys that are usually regularly conducted worldwide, in order to document the economic and nutritional importance of food consumption away from home. In each country, two survey modules, one long (100 food items) and one short (30 food items) were developed, to measure individual-level food consumption away from home over the last 7 days. The modules were relatively validated in comparison with data from three non-consecutive 24-hour dietary recalls conducted over the same 7-days period. The validation was conducted in different settings (urban, peri-urban and rural) in Burkina Faso (n = 924) and Vietnam (n = 918). In both countries, a good concordance between the 24-hour dietary recalls and the modules in their ability to identify a person as having consumed food away from home (>77%) was found. However, in both countries, both modules underestimate the mean energy intake coming from foods consumed away from home (from 122 to 408 kcal) while they overestimate the budget allocated to it (from -0.07 to -0.29 USD/day). None of developed food away from home modules were considered as valid. There is a need for the international community to continue to work on developing and validating tools capable to estimate nutritional intakes related to food

personal. In the protocol approved by both ethics board, there is nothing stating that the data will be made publicly available. In the case of Vietnam the data are owned by the country and permission from the Ministry of Health is required for free and public access to the data. Furthermore, in the information letter given to each partipants it is stated: « The information you provide will not be divulged: only the people involved in this survey will have access to the information collected, anonymously. » In Burkina Faso the person to contact to get access to the data is: Mr Yves Gérard Bazié, Head of the Studies, Forecasting and Policy Department at the Permanent Secretariat for Agricultural Sector Policies, Ouaggadougou, Burkina Faso. Mr Bazié was responsible for submitting the protocole to the ethic committee in Burkina Faso and guaranteed that the data collection followed the protocol submitted to the ethics committee. His contact is: sevy_baz@yahoo.fr In Vietnam the person to contact to get access to the data is: Huynh Nam Phuong from the National Institute of Nutrition, based in Hanoi, Vietnam. Huynh Nam Phuong is the Head of Science Management Section of National Institute of Nutrition and she was involved in data management. Her contact is: quanlykhoahoc.ninvn@gmail.com.

**Funding:** This work was supported by the Innovative Metrics and Methods for Agriculture and Nutrition (IMMANA) under Grant 16218321. The sponsor did not play any role in the study design, data collection and analysis, decision to publish or preparation of the manuscript.

**Competing interests:** The authors have declared that no competing interests exist.

consumption away from home and that could be added to regular national household-level surveys.

## 1. Introduction

The consumption of food away from home is increasing rapidly at a global scale, becoming one of the major symbols of rapidly transitioning food systems. Defined as "food items that are obtained and consumed from restaurants, cafeterias, food trucks, street outlets, relatives homes, or vending machines" [1], food consumed away from home is steadily taking up a larger share of household food expenditures worldwide [2–4]. Rising incomes and urbanization, increased participation of the female workforce in the formal economy, socio-cultural changes and modifications to the food environment, are the main drivers of this rapid increase in food consumed away from home [2]. In high-income countries, it has been shown that food consumed away from home was associated with unfavorable nutritional outcomes such as higher energy intake and poorer quality diets [5,6], as well as higher risk of being obese or overweight [6–8]. In low- and middle-income countries (LMICs), the phenomenon is less well documented. However, the limited existing empirical evidences suggest that food consumed away from home has an adverse impact on diet quality, nutritional status and health outcomes also in LMICs [9].

Individual-level dietary surveys are the most accurate source of information to estimate the contribution of foods consumed away from home to diet quality and its consequences on health. Unfortunately, collecting and processing individual dietary data on a large, representative scale requires infrastructure, capacities and important financial resources that are often lacking in LMICs [10], leading to a scarcity of data in these countries [9]. Conversely, Household Consumption and Expenditure Surveys (HCES) are conducted regularly in most LMICs and are sometimes used to derived food consumption data [11–13]. These surveys could therefore provide an opportunity to better document food consumed away from home. However, in their current forms, HCESs are not detailed enough to do so. A 2014 review, focusing on the reliability and relevance of food data collected in national HCESs, reported that amongst over 100 HCESs that documented food consumed away from home in LMICs, more than one-third used only one question in the entire questionnaire to do so [14]. The same review reported that the data on food consumed away from home were collected through questions at the individual level in only 17 surveys [14]. However, even for these questions at individual level, the amount of information collected remains minimal. For example, the module used in the 2018/19 Harmonized Survey on Households Living Standards in Burkina Faso only collects information for 8 items highly heterogeneous in terms of nutrient composition [15].

This gap in the ability to correctly measure food consumed away from home in LMICs raises serious concern. If left unaddressed, it could potentially result in inadequate policies and interventions with important unintended consequences for the nutritional and health status of the hundreds of millions of individuals who consume food away from home every day, either at work (from, e.g., canteen, street vendors or vending machines) or for leisure (from e.g. restaurants and fast food). The objective of the present paper is to report the key findings of an international initiative which aimed at developing, field-testing and validating specific individual-level survey modules with the ambition to add these modules to HCES questionnaire current used in LMICs. The modules were designed to document the economic and nutritional importance of the consumption of foods and beverages both obtained and consumed away

from home. The modules were tested in Vietnam and Burkina Faso and relatively validated against three non-consecutive 24-hour dietary recalls.

## 2. Methods

We build upon an experimental method where the information on food consumed away from home obtained from two specific survey modules we developed over a one-week period is compared against the information on food consumed away from home extracted from three repeated 24 hours dietary recalls over the same week period. Individual respondents are randomly assigned to one of the two modules developed.

### 2.1. Development of two food consumed away from home modules

Based on previous studies [16,17] and local partners' expertise, a long and a short version of an individual-level module were developed by the research team. The long version of the module (referred to as 'long-list module' in the rest of the article) was designed to contain around 100 food items that could be consumed away from home in each studied country. The shorter version of the module (referred to as 'short-list module' in the rest of the article) contained around 30 food items and was obtained by eliminating some food items from the long-list which consumption was deemed extremely rare by local partners, and grouping other food items into more generic food item groups based on the similarity of their nutritional composition (S1 Table 1 in S1 File and Table 2). For instance, dishes made of different vegetable basis without differentiating between vegetables were grouped together, keeping only the differentiation between vegetarian, meat and fish options.

The long- and short-list modules were designed as 7-day recall, that is, for each food item, respondents were asked (i) whether the food item was consumed or not during the last 7 days and if so (ii) how many times, (iii) how many times they had to pay for it, (iv) what was the total amount of money they spent on it over the last 7 days, (v) what was its main consumption place and, if different, (vi) what was its main procurement place.

### 2.2. Sample size

For the validation objective, the maximum acceptable difference between the tested modules and the reference (24-hour dietary recalls) was considered to be 20%. In the absence of any food consumption survey data available on out-of-home consumption, we have only been able to make this choice on the basis of budgetary data. Based on the results from a previous national study on household living standards conducted in 2018 in Vietnam [18], the average individual weekly budget related to the away from home food consumption was estimated to be about 2.5 USD, which gave a maximum acceptable difference of 0.5 USD. It was assumed that the standard deviation of the difference between the tested module and the reference module could be around to 0.20. To validate a module with the Bland & Altman method, for a power of 80%, if the expected standard deviation of the difference was set to 0.22, a mean difference close to 0 (0.001) and a confidence level of 95% for the limits of agreement [19], 317 households should be included. If 25% were to be added to take into account potential drop-outs, it means that at least 422 households should be surveyed. This number was rounded to 450 to anticipate a higher rate of drop-outs and /or an underestimated standard deviation. In Vietnam 450 households per module were therefore included, which made a total sample of 900 households (450 households for each module). In Burkina Faso, the same sampling calculation method was applied using data from a previous national HCES conducted in 2018–2019, where individual weekly budgets for food consumed away from home were 1.26 USD on

average [15] and led to a sample size of 300 households per modules which made a total of 600 households.

## 2.3. Study participants

Within each household, individuals aged 18–59 years were randomly selected to be surveyed. In Vietnam, one adult per household was selected whereas in Burkina Faso one adult and potentially his/her partner were selected. This difference in approach was due to the fact that the intra-household variation was of interest and that it was only possible to collect these additional data in Burkina Faso because of budget limitation in Vietnam. The interviews were conducted in national languages (Vietnamese in Vietnam, and Moore or French in Burkina Faso) by trained enumerators. The protocol was developed according to the guidelines in the Declaration of Helsinki and validated by national Ethics Committees (Ethics council in biomedical research of the National Institute of Nutrition, Vietnam, approval number 815/VDD-QLKH, October 12, 2021 and the "Comité Ethique pour la Recherche en Santé" in Burkina Faso, approval number 2022-02-022). Written informed consent was obtained from each participant. In Vietnam, based on the latest population census (2019), a random sample of 918 participants was recruited from Cau Giay District of Hanoi (urban), Gia Lam District of Hanoi (peri-urban) and Vu Thu District of Thai Binh (rural). In Burkina Faso, a random sample of 924 participants was recruited from Ouagadougou (urban), the Great Ouagadougou (peri-urban) and Moackin (rural) on the basis of a stratification made from the latest large scale representative survey (Enquête Harmonisée sur le Conditions de Vie des Ménages) available [15].

## 2.4. Study design

The two versions of food consumed away from home modules (long and short) were relatively validated against data obtained from three non-consecutive 24-hour dietary recalls (two weekdays and one weekend day) conducted with the same participants over the same 7-days period. In Vietnam, each participant was randomly assigned to one of the two modules, resulting in 477 individuals assigned to the short-list module and 441 participants assigned to the long-list module. In Burkina Faso, in the same way, each individual was randomly assigned to one module, with the differences that in case the individual was living as a couple, her/his partner was potentially surveyed as well. This resulted in a total of 460 participants testing the short-list module and 464 participants testing the long-list module.

In both countries, once participants were recruited, trained enumerators explained the aims and the procedures of the study, assessed whether participants met the inclusion criteria and collected the informed consent during the first interview. Over the week, three appointments were made with the surveyed individuals. On the first appointment, enumerators collected data on sociodemographic information (age, gender, household size, marital status, educational level, main activity and socio-economic situation) and filled the first 24-hour dietary recall. On the second appointment, enumerators filled another 24-hour dietary recall. On the third appointment, enumerators filled the last 24-hour dietary recall as well as either the short-list or the long-list module depending, on which subgroup the individual respondent had been assigned to.

## 2.5. Data collection

Data were collected on tablets using the mobile data collection application INDDEX24 created by the INDDEX Project [16,17], to which the two long list and short list modules were added. In Vietnam data were collected from June to July 2022 and in Burkina Faso data were collected from May to July 2022.

**2.5.1. Repeated 24-hour dietary recalls.** The detailed procedure of conducting 24-hour dietary recalls using INDDEX24 has been published elsewhere [16,17]. Briefly, enumerators followed the multipass method, which includes 4 passes. During the first pass, they asked for a quick list of foods, time and place of consumption. In the second pass, enumerators recorded the full list of all foods, beverages and mixed dishes recalled as consumed by the respondents during the previous 24 hours. For each food item, beverage and mixed dish, respondents were asked about the place of consumption and procurement. The amount consumed by the participants was estimated in the third pass, using the portion size estimation methods integrated in INDDEX24: direct weight, life-sized photos, or proxy weight using either dried sorghum (in Burkina Faso), water, or price and standard units. When the food item, beverage or mixed dish was both obtained and consumed away from home, participants were asked in the third pass about the amount of money spent. Enumerators checked the summary of the interview in the fourth pass.

## 2.6. Assessment of nutrient intakes

**2.6.1. Assessment based on repeated 24-hour dietary recalls.** To convert the food intakes collected with the three 24-hour dietary recalls into energy and macronutrient intakes, we used the relevant portion conversions, yield factors, retention factors and nutritional compositions for Vietnam and Burkina Faso compiled previously for other studies using the INDDEX24 tool [16,17]. When computing energy and macronutrients intakes over a week, a weighing average system between the two weekdays and the weekend day was used to better reflect the weekly consumption (each week day was weighted 2.5 and the weekend day was weighted 2) and these values were expressed in daily equivalent (by dividing by 7).

**2.6.2. Assessment based on food consumed away from home modules.** To convert the food frequency data collected with the modules into food intakes and then into energy and macronutrient intakes, the following process was followed. First, relevant food items, beverages and mixed dishes from repeated 24-hour dietary recalls were matched with each item from the modules. Then, average portion sizes for food items, beverages and mixed dishes as well as nutrient composition for each item of the modules were calculated based on data from the 24-hour dietary recalls. The daily intake of each item was calculated by multiplying the number of eating events by the average portion size and dividing by seven. The daily energy and macronutrient intakes of each item were calculated by multiplying the daily intake by the average nutrient composition expressed per 100g.

## 2.7. Assessment of expenditures

The local currencies were converted to USD at the conversion rates corresponding to the survey periods. Monetary expenditure from the repeated 24-hour dietary recalls was computed using the same average weighting system between the two weekdays and the weekend day described previously to account for intra-week variation and these values were expressed in daily equivalent. To calculate the monetary expenditure collected from the food consumed away from home modules, the total amount was divided by seven.

## 2.8. Statistical analyses

In each country, modules-based samples were tested for differences (using t-test or Chi2 test). For testing the validity of the developed module, first, the percentage of participants who reported consumption away from home in both the modules and the three 24-hour dietary recalls was computed (and referred to as the concordance). For the relative validity of the modules compared to the three 24-hour dietary recalls, Spearman correlation coefficients were

calculated to estimate the strength and direction of association for energy and macronutrient intakes, as well as monetary expenditure, as estimated from the three 24-hour dietary recalls and the modules. The Wilcoxon signed-rank test was also used to test whether the energy and the macronutrient intakes and monetary expenditure differed between the three 24-hour dietary recalls and the developed modules. Finally, Bland and Altman analysis [20,21] was used to graphically evaluate the agreement between the three 24-hour dietary recalls and the modules by plotting the mean differences between the two methods against the average estimation with the 95% limit of agreement calculated as the mean difference ±1.96. The level of significance used was $p < 0.05$. All analyses were conducted using Stata 17 (Statacorp, College Station, TX).

## 3. Results

### 3.1. Study participants

The characteristics of the participants are displayed in Table 1. In both countries, the mean age of the respondents was comparable (37.1 in Vietnam and 35.6 in Burkina Faso). The samples were slightly skewed in favour of women (54.9% in Vietnam and 57.0% in Burkina Faso). The most notable differences between the samples from the two countries were (i) the level of education–as about four out of ten adults in Burkina Faso had no education, whereas in Vietnam around three-quarters of participants had at least upper secondary education–and (ii) the marital status–as a larger proportion of participants in Burkina Faso were married compared to Vietnam (82.4% and 63.2%, respectively). There were some significant differences between the short and long-list samples for age and education in Vietnam. There were no significant differences in Burkina Faso between the two samples.

**Table 1. Descriptive characteristics of the samples.**

| | Vietnam | | | | Burkina Faso | | | |
|---|---|---|---|---|---|---|---|---|
| | All sample | Short-list module | Long-list module | | All sample | Short-list module | Long-list module | |
| | (n = 918) | (n = 477) | (n = 441) | p value | (n = 924) | (n = 460) | (n = 464) | p value |
| **Mean age** (± sd) * | 37.1 (±13.6) | 38.7 (±13.6) | 35.3 (±13.3) | <0.05 | 35.9 (±10.4) | 35.6 (±10.7) | 35.6 (±10.7) | n.s |
| **Gender (%)** | | | | | | | | |
| Male | 45.10% | 43.00% | 47.40% | n.s | 43.00% | 43.50% | 42.50% | n.s |
| Female | 54.90% | 57.00% | 52.60% | | 57.00% | 56.50% | 57.50% | |
| **Living area (%)** | | | | | | | | |
| Urban | 51.40% | 49.30% | 53.70% | n.s | 52.30% | 53.50% | 51.50% | n.s |
| Peri-urban | 32.00% | 32.70% | 31.30% | | 30.80% | 28.90% | 32.80% | |
| Rural | 16.60% | 18.00% | 15.00% | | 16.90% | 17.60% | 16.20% | |
| **Educational level (%)** | | | | | | | | |
| No education | 0.20% | 0.40% | 0.00% | <0.0001 | 38.80% | 39.50% | 37.70% | n.s |
| Up to lower secondary education | 23.90% | 29.40% | 17.90% | | 42.00% | 43.20% | 40.70% | |
| At least upper secondary education | 75.90% | 70.20% | 82.10% | | 19.40% | 21.60% | 11.50% | |
| **Marital status (%)** | | | | | | | | |
| Married, living with a partner | 63.20% | 66.70% | 59.40% | n.s | 82.40% | 80.20% | 84.50% | n.s |
| Widowed, divorced, separated | 3.90% | 4.00% | 3.90% | | 2.60% | 2.80% | 2.40% | |
| Never married | 32.90% | 29.40% | 36.70% | | 15.00% | 17.00% | 13.10% | |

sd: standard deviation; n.s: non-significant.

* n = 915, 3 missing values for age.

## 3.2. Food consumption away from home

The 24-hour dietary recall data indicate that about two-thirds of the respondents in Vietnam consumed at least one food item away from home in the week (64.7%) while eight out of ten respondents do in Burkina Faso (79.8%). In comparison, the proportion of respondents who reported to consume away from home based on the module data was slightly higher: 68.1% for the short-list module and 70.7% for the long-list module in Vietnam, and 89.7% for the short-list module and 88.7% for the long-list module in Burkina Faso. The mean energy intake coming from foods consumed away from home was much higher in Burkina Faso compared to Vietnam, and so were protein, carbohydrate and fat intakes (see next section).

## 3.3. Validity of the modules

In both Vietnam and Burkina Faso, the concordance was slightly higher for the long-list module than for the short list one: 77.5% for the short-list and 78.9% for the long list in Vietnam, and 84.1% and 85.1% for the short-list and the long-list in Burkina Faso (**Table 2**). In Vietnam, around 13.0% of the participants reported to have consumed food away from home based on the modules but failed to do so in the 24-hour dietary recalls. In Burkina Faso this proportion was about 12% of the participants (**Table 2**). Likewise, 9.5% and 7.5% of participants in Vietnam reported to have consumed food away from home on the 24-hour dietary recalls but failed to do so in the modules, respectively the short- and the long-lists modules. This misclassification was lower in Burkina Faso, respectively 3.1% and 2.6% for the short- and the long-lists modules (**Table 2**).

**3.3.1. Nutritional validity.** In Vietnam, as well as in Burkina Faso, compared to the 24-hour dietary recalls, the two modules underestimate the mean energy intake coming from foods consumed away from home (**Table 3**). In both countries, the underestimation is greater with the short-list module with a mean difference of 154 kcal/day (-56%) in Vietnam, and a mean difference of 422 kcal/day (-48%) in Burkina Faso, while the mean difference evaluated for the long list modules is 122 kcal/day (-42%) in Vietnam and 408 kcal/day (-50%) in Burkina Faso (**Table 3**). For macronutrients, the results are similar i.e. the modules tend to underestimate protein, carbohydrate and fat intake.

Overall, in both countries, there is systematic positive and moderate Spearman correlations between the estimation from the modules and from the 24-hour dietary recalls ranging from 0.51 to 0.55 for the short-list module and from 0.57 to 0.60 for the long-list module in Vietnam, and from 0.47 to 0.58 for the short-list module and 0.50 to 0.59 for the long-list module in Burkina Faso (**Table 3**). In both countries the mean intakes coming from food consumed away from home and estimated from the modules for energy, proteins, carbohydrates and fats

**Table 2. Concordance of the prevalence of consumption away from home according to the modules and the repeated 24-hour dietary recalls.**

|  | Vietnam | | Burkina Faso | |
|---|---|---|---|---|
|  | **Short-list (n = 477)** | **Long-list (n = 441)** | **Short-list (n = 460)** | **Long-list (n = 464)** |
| Consumers in both module & 24HDR* | 55.1% | 57.1% | 76.9% | 76.0% |
| Non-consumers in both module & 24HDR | 22.4% | 21.8% | 7.2% | 9.1% |
| **Total concordance** | 77,5% | 78.9% | 84.1% | 85.1% |
| Consumers in module but not 24HDR | 13.0% | 13.6% | 12.8% | 12.3% |
| Consumers in 24HDR but not module | 9.5% | 7.5% | 3.1% | 2.6% |
| **Total non-concordance** | 22.5% | 21.1% | 15.9% | 14.9% |

* 24-hour dietary recall.

**Table 3. Relative validity of the food consumed away from home modules compared with the 3 repeated 24-hour dietary recalls.**

| | | Short-list module | | | | | Long-list module | | | | |
|---|---|---|---|---|---|---|---|---|---|---|---|
| | | 3* 24-hour dietary recalls | Module | Mean difference* | Spearman rank correlation | Wilcoxon signed rank test | 3* 24-hour dietary recalls | Module | Mean difference* | Spearman rank correlation | Wilcoxon signed rank test |
| Vietnam | Mean energy intake (kcal/d) | 272 | 118 | 154 | 0.53 | <0.0001 | 290 | 169 | 122 | 0.60 | <0.0001 |
| | Mean protein intake (g/d) | 12.1 | 4.8 | 7.3 | 0.55 | <0.0001 | 12.9 | 6.5 | 6.5 | 0.59 | <0.0001 |
| | Mean carbohydrates intake (g/d) | 35.6 | 16.3 | 19.3 | 0.51 | <0.0001 | 37.9 | 23.9 | 14 | 0.60 | <0.0001 |
| | Mean fat intake (g/d) | 9.1 | 3.7 | 5.4 | 0.55 | <0.0001 | 9.9 | 5.2 | 4.7 | 0.57 | <0.0001 |
| | Mean monetary expenditure (USD/d) | 0.35 | 0.51 | -0.16 | 0.56 | 0.0009 | 0.37 | 0.66 | -0.29 | 0.60 | <0.0001 |
| Burkina Faso | Mean energy intake (kcal/d) | 873 | 451 | 422 | 0.58 | <0.0001 | 817 | 409 | 408 | 0.59 | <0.0001 |
| | Mean protein intake (g/d) | 28.1 | 13.0 | 15.1 | 0.54 | <0.0001 | 26.0 | 11.7 | 14.3 | 0.56 | <0.0001 |
| | Mean carbohydrates intake (g/d) | 123.4 | 56.4 | 67.0 | 0.57 | <0.0001 | 119.1 | 52.4 | 66.7 | 0.57 | <0.0001 |
| | Mean fat intake (g/d) | 24.2 | 14.1 | 10.1 | 0.47 | <0.0001 | 21.4 | 12.7 | 8.7 | 0.50 | <0.0001 |
| | Mean monetary expenditure (USD/d) | 0.39 | 0.47 | -0.07 | 0.73 | <0.0001 | 0.33 | 0.49 | -0.16 | 0.70 | <0.0001 |

* Mean difference = data from average 3 24-hour dietary recalls-data from module.

differs significantly from what was estimated from the 24-hour dietary recalls (all p-values <0.0001) (**Table 3**).

The Bland and Altman plots, visually confirms the underestimation of energy intakes by the modules. In both countries, the V-shape of the distribution indicates that the higher the values of energy intakes from food consumed away from home, the higher the misestimating between the modules and the repeated 24-hour dietary recalls (**Fig 1**).

The Bland and Altman plots for proteins, carbohydrates and fats intakes show similar patterns to the ones for energy, and therefore confirm the overall underestimation of intakes by the modules (**S1 Fig 1 in S1 File and Fig 2**).

**3.3.2. Budgetary validity.** In both countries, compared to the 24-hour dietary recalls, the short list and long list modules overestimate the budget allocated to food consumed away from home (**Table 3**). In both countries the overestimation is greater with the long-list module compared to the short-list module: -0.29 USD/day (+78%) and -0.16 USD/day (+46%), respectively in Vietnam; -0.16 USD/day (+48%) and -0.07 USD/day (+18%), respectively in Burkina Faso. In Burkina Faso, the Spearman correlations between the estimation from the modules and the 24-hour dietary recalls are slightly better compared to Vietnam (0.73 vs 0.56 for the short-list and 0.70 vs 0.60 for the long-list) (**Table 3**). In both countries the mean budget allocated to food consumed away from home estimated from the modules differs significantly from what was estimated from the average three 24-hour dietary recalls (all p-values <0.001) (**Table 3**).

As for energy and macronutrients, the V-shape of the distribution indicates that the greater the budget allocated to food consumed away from home, the greater the misestimation

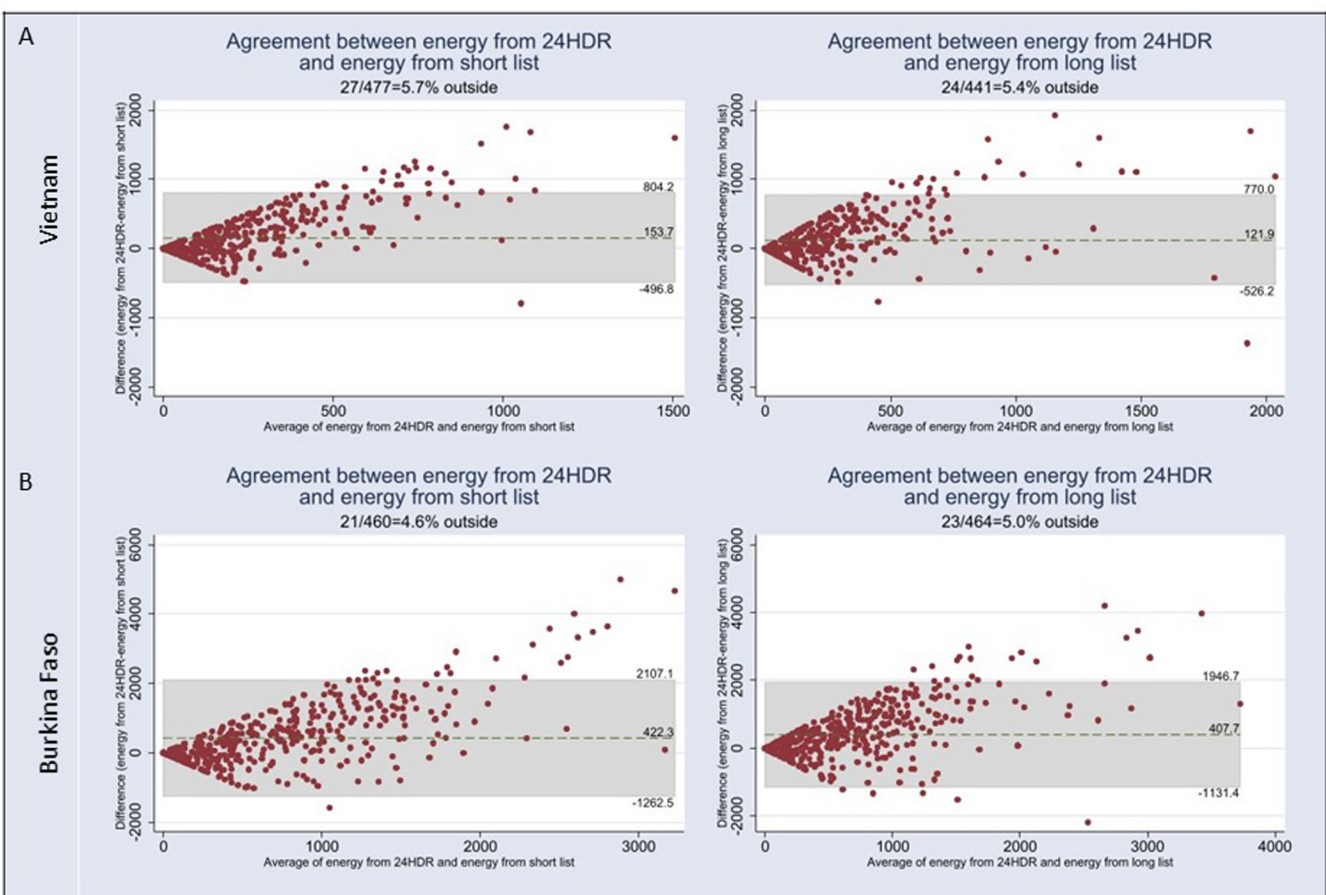

**Fig 1.** Bland and Altman plots for energy–A) Vietnam; B) Burkina Faso. 24HDR: 24-hour dietary recall.

between the modules and the repeated 24-hour dietary recalls (**Fig 2**). The limits of agreement are wider in Vietnam than to Burkina Faso, which is consistent with the conclusion drawn from the correlation coefficients.

## 4. Discussion

Several studies conducted previously have compared estimates derived from HCES and individual dietary intakes but without a specific focus on food consumption away from home [12,13]. A systematic review on the same topic reported that overall dietary intakes derived from HCES tended to be oversestimated [22].

Even though there have been some intitiatives [23] and international recommendations [24] to improve the assessment of food consumption away from home within the HCES, to the best of our knowledge, this paper reports the first attempt to develop and validate specific survey modules that could have been added to HCES to better estimate the nutritional and economical aspects of the rapidly developing food consumption away from home behaviour. Although we found differences in performance between modules and countries, both modules, in both countries, significantly underestimated energy and macronutrient intakes, but overestimated budget allocated to food consumed away from home. Based on the statistical analyses conducted our study does not allow us to conclude on the validity of the modules, either to measure energy and macronutrients intakes or expenditures, and other tests should be carried out,.

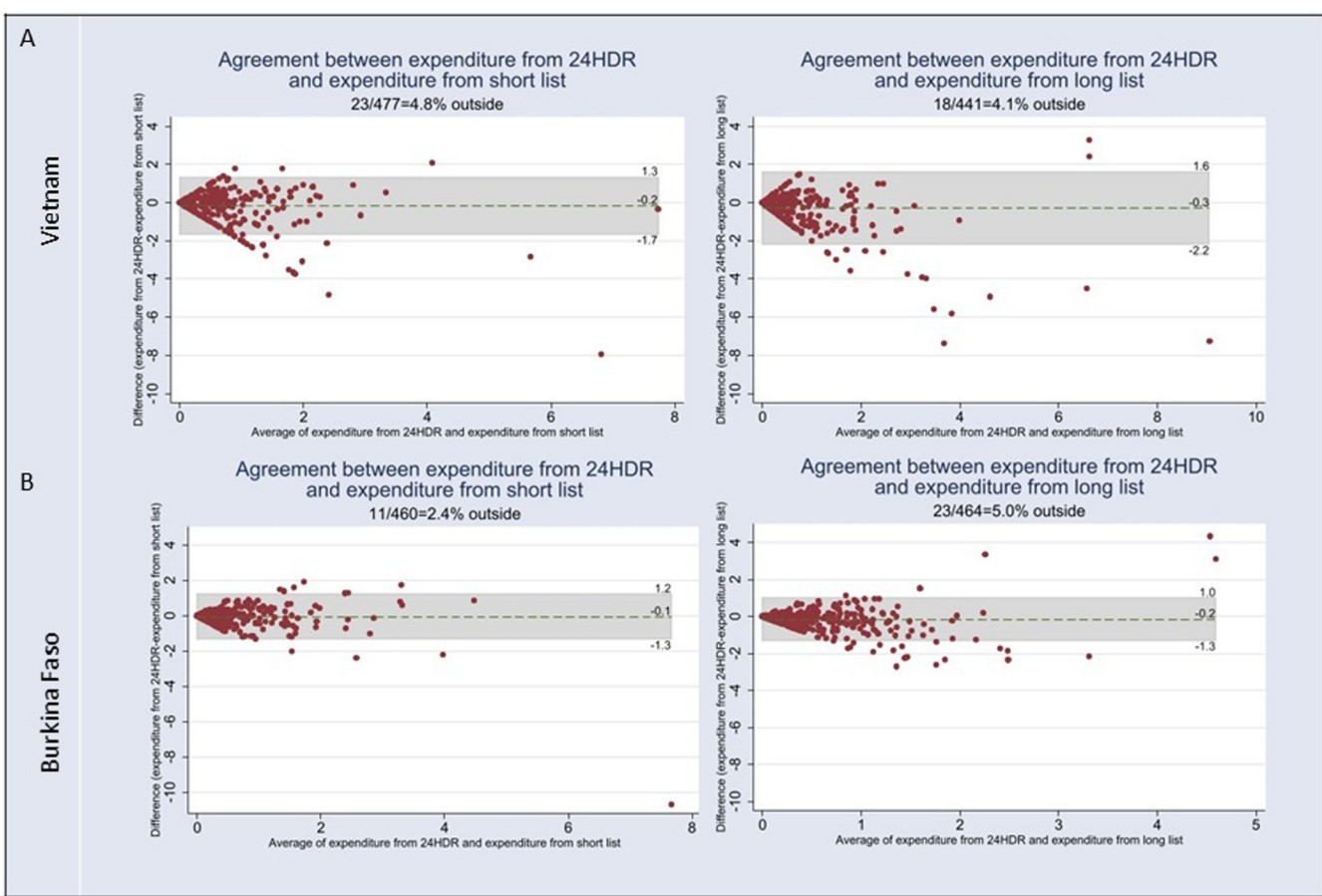

**Fig 2. Bland and Altman plots for monetary expenditure.** A) Vietnam; B) Burkina Faso. 24HDR: 24-hour dietary recall.

Overall, the long-list modules performed slightly better than the short-list modules to classify participants in right category of a consumer of food away from home, and the modules worked better in Burkina Faso compared to Vietnam. This might be explained by the fact that the long-list modules were much more detailed compared to the short-list ones (100 *vs.* 30 items) hence leading potentially to a larger number of food items omitted with the short-list module, or to difficulties of respondents to identify the specific food items consumed in the short list modules because of the regroupings made. The long-list module estimation was closer to the one from repeated 24-hour dietary recalls compared to the short-list module. There was a systematic positive and moderate correlation between the estimation from the modules and the repeated 24-hour dietary recalls. Overall the Spearman correlation coefficients were slightly better for the long-list module compared to the short-list module. This is consistent with what is usually reported in the literature where questionnaires with higher number of food items generally have higher correlation coefficients compared to those with smaller number of food items [25].

The V-shape of the Bland and Altman plots indicated a bias in the modules suggesting that the higher the values of intakes or monetary expenditures, the greater the misestimation between the modules and the repeated 24-hour dietary recalls [26].

Few studies conducted either in Vietnam or in Burkina Faso reported data on food away from home behavior. In a study conducted in 2006–2007 amongst Vietnamese adolescents,

Lachat et al. [27] reported that 73% of the adolescents surveyed had at least one away from home consumption occasion over a week. In another study conducted more recently in Northen Vietnam amongst adults, Sonneveld [28] found a gradient of away from home consumption per week ranging from 12% in rural areas to 62% in urban areas. Although conducted in Vietnam, these two studies were implemented on a different population or in different location, but also before COVID-19 making the comparison with the results from the present study difficult. It is plausible than the prevalence of food consumed away from home was lower in Vietnam at the time of the survey because the closure of restaurants had been released recently and individuals may be fearing contamination.

Only one study conducted in Burkina Faso was found [29] but was conducted in urban area only, on women of reproductive age and considered food away from home from a meal perspective, once again making the comparison with the present study difficult.

The main limitation of this study is the use of the repeated 24-hour dietary recalls as a reference method for the validation of the developed modules. Even though away from home consumption is a quite common behavior [2], for some individuals, it can still be an occasional behaviour. For example, in studies investigating the consumption over one week or more, it was found that 50.0% of Turkish adults aged 18–30 years consumed food away from home 1–2 times per week on week days [30] and 24.9% of Ethiopian adults ($\geq$18 years) consumed food away from home once a week [31]. It is therefore questionable to only use three 24-hour dietary recalls to capture a behaviour that may occur occasionally over the course of a week, as we may miss specific events. An alternative would have been to use a 7-day food diary as a reference method. However, it has been shown that diary-keeping surveys are burdensome for respondents, potentially leading to bias from diary fatigue, and there are many benefits of recall over diary data collection for food items [32]. Furthermore, this appraoch would have been very difficult to implement in Burkina Faso, given the high prevalence of illiteracy in our sample.

In the present study, in slightly more than 10% in both countries, the modules captured away from home consumption when there was no recording of food consumed away from home during the three repeated 24-hour dietary recalls. This indicates that the three 24-hour recalls implemented over the week are not frequent enough to record every consumption of food away from home instances and that some of these consumptions of food away from home instances occur during the days that are not covered by the 24-hour recalls. This is not completely unexpected since only two of the three 24-recalls were taking place during the week days (and the third during the week-end). One potential way to reduce the risk of missing some consumption of food away from home instances would be to to use a 7-day food diary or daily weighed food records over seven days. These methods however generate a bigger burden for participants and require greater compliance compared to repeated 24-hour dietary recalls. Additionnaly, they could lead the participants to modify thier diets (especially the away from home consumption) to make it easier to record [33].

What is more difficult to explain in this context are the cases where participants reported consuming food away from home in the 24-hour dietary recalls but failed to do so in the 7-day recall modules. The proportions are lower but not negligible, especially in Vietnam (more than 9%). In this case, a possible explanation relates to the memory bias that may increase with the length of the recall period (in the present case one day vs. one week).

Another limitation is that while aiming at calculating mean portion size for food items in the modules, as there was no previous individual dietary data available in both countries, data from the repeated 24-hour dietary recalls were used. However, for certain food items, occasions of consumption away from home were limited. For these items, the mean portion sizes were therefore computed regardless to the place of consumption even though some studies reported that portion sizes of food away from home tended to be bigger [34,35]. However,

when possible, portion sizes between the same foods consumed away from home and those consumed at home were compared, and the differences were modest.

Finally, one other weakness of the present approach is to consider 24-hour dietary recalls as a benchmark to record expenses related to food consumed away from home, especially in the case of the use of the INDDEX24 tool where no expenses are reported and we had to include a non-mandatory comment section to record it. This may have jeopardized the quality of budget information and this may explain the underestimation of expenses with the 24-hour dietary recalls relative to the developed modules who may provide more accurate information. Additionally, different approaches were used to evaluate the nutritional aspects of food consumed away from home and the economic aspects. For the nutritional aspect, the number of times people reported to have eaten away from home was collected and then multiplied by an average portion size, whereas for the economic aspect, an estimate of the total budget was collected. A more harmonized approach could produce more consistent results.

## 5. Conclusion

In a context of a worldwide increase of food away from home consumption and a scarcity of national individual-level dietary surveys in LMICs [9], there is a need for the international community to continue to work on developing and validating tools capable to estimate nutritional intakes related to this particular behavior and that could be added to regular national level conducted survey such as HCES.

## Supporting information

**S1 Checklist.**
(DOCX)

**S1 File.** S1 Table 1. **Long list and short list, Vietnam**. S1 Table 2. **Long list and short list, Burkina Faso**. S1 Fig 1. **Bland and Altman plots for Vietnam: A) for protein; B) for carbohydrate; C) for lipid**. 24HDR: 24-hour dietary recall. S1 Fig 2. **Bland and Altman plots for Burkina Faso: A) for protein; B) for carbohydrate; C) for lipid**. 24HDR: 24-hour dietary recall.
(DOCX)

## Acknowledgments

We would like to think all the participants of the study as well as Nathalie Troubat and Desiree Lucassen who participated in October 2023 to a workshop for discussing the results of the study.

## Author Contributions

**Conceptualization:** Edwige Landais, Elodie Maître d'Hôtel, Mai Truong Tuyet, Christophe Béné, Eric O. Verger.

**Data curation:** Raphaël Pelloquin, Yen Bui Thi Thao, Trang Tran Thi Thu.

**Formal analysis:** Edwige Landais, Raphaël Pelloquin, Elodie Maître d'Hôtel, Eric O. Verger.

**Funding acquisition:** Edwige Landais, Elodie Maître d'Hôtel, Christophe Béné, Eric O. Verger.

**Investigation:** Edwige Landais, Elodie Maître d'Hôtel, Mai Truong Tuyet, Nga Hoang Thu, Yen Bui Thi Thao, Ha Do Thi Phuong, Trang Tran Thi Thu, Jérôme Somé, Eric O. Verger.

**Methodology:** Edwige Landais, Elodie Maître d'Hôtel, Mai Truong Tuyet, Nga Hoang Thu, Ha Do Thi Phuong, Jérôme Somé, Christophe Béné, Eric O. Verger.

**Project administration:** Elodie Maître d'Hôtel.

**Resources:** Mai Truong Tuyet.

**Software:** Nga Hoang Thu, Jérôme Somé.

**Supervision:** Edwige Landais, Elodie Maître d'Hôtel, Mai Truong Tuyet, Nga Hoang Thu, Yen Bui Thi Thao, Trang Tran Thi Thu, Jérôme Somé.

**Validation:** Edwige Landais, Raphaël Pelloquin, Yen Bui Thi Thao, Eric O. Verger.

**Writing – original draft:** Edwige Landais, Christophe Béné, Eric O. Verger.

**Writing – review & editing:** Raphaël Pelloquin, Elodie Maître d'Hôtel, Mai Truong Tuyet, Nga Hoang Thu, Yen Bui Thi Thao, Ha Do Thi Phuong, Trang Tran Thi Thu, Jérôme Somé.

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
