## [Decision Letter · Decision Letter 0]

29 Aug 2024

PONE-D-24-21954Assessing food consumed away from home in low-and middle-income countries by developing specific modules for household surveys: experimental evidence from Vietnam and Burkina FasoPLOS ONE

Dear Dr. Landais,

Thank you for submitting your manuscript to PLOS ONE. After careful consideration, we feel that it has merit but does not fully meet PLOS ONE’s publication criteria as it currently stands. Therefore, we invite you to submit a revised version of the manuscript that addresses the points raised during the review process.

Both reviewers feel that the article is publishable, but have made some minor constructive suggestions. Please take these into account.

We look forward to receiving your revised manuscript.

Kind regards,

Susan Horton

Academic Editor

PLOS ONE

Journal Requirements:

4. In the online submission form, you indicated that "Data described in the manuscript, code book, and analytic code will be made available upon request pending application and approval by the authors of the current study."

Additional Editor Comments:

Please see the helpful and constructive suggestions of the two reviewers

Reviewers' comments:

Reviewer's Responses to Questions

**Comments to the Author**

1. Is the manuscript technically sound, and do the data support the conclusions?

Reviewer #1: Partly

Reviewer #2: Yes

2. Has the statistical analysis been performed appropriately and rigorously? 

Reviewer #1: Yes

Reviewer #2: Yes

3. Have the authors made all data underlying the findings in their manuscript fully available?

Reviewer #1: Yes

Reviewer #2: Yes

4. Is the manuscript presented in an intelligible fashion and written in standard English?

Reviewer #1: Yes

Reviewer #2: Yes

5. Review Comments to the Author

Reviewer #1: I generally have a favorable impression of this manuscript. The author(s) do a good job of motivating the need to develop measures of dietary intake from food away from home in low- and middle-income countries using household consumption and expenditure surveys. The design of two survey modules (long- and short-form food frequency questionnaires) to be used to approximate dietary intake in household consumption and expenditure surveys, sample design and recruitment, and administration of both the survey modules and 24-hour recalls are described well and seem mostly appropriate although I have some clarifying questions on their approach described below. Below are more detailed comments for the author(s) to consider.

1. Its not surprising that the weighted average 24-hour recall estimates of energy and macronutrient intake are almost double that of what is reported in a 7-day recall food frequency questionnaire. The choice of 7-day recall versus 24-hour recall seems a bit of an apples to oranges comparison and I’m glad that the author(s) examine this a bit in the Discussion section. But I wonder why the author(s) chose a 7 day recall rather than a diary approach with their survey modules? Also, it seems like quite a burden to collect both the 24-hour recall and food frequency survey modules in the 3rd visit. I wonder how that affected the responses to the food frequency questionnaire and the recall.

2. The author(s) do a pretty good job of summarizing the literature but it seems like there have been a handful of studies not included that have examined the issue of validating household consumption and expenditure data in measuring dietary intake (e.g., Coates et al. 2017; Sununtnasuk and Fiedler 2017; Karageorgou et al 2019; tang et al. 2022). I encourage the author(s) to look at these articles as it seems like many of the points raised in the discussion overlap with the central points of these articles.

3. The author(s) conclude in the abstract and introduction that “None of the developed food away from home modules were considered valid,” which is a bit too strong. The measures were only validated using static measures, i.e., mean comparisons in one time period. However, I wonder how well the survey modules in measuring changes in energy and nutrient composition of food away from home consumption over time? It seems to me that usually we are interested in capturing changes in energy and nutrient consumption. Also, is heterogeneity in concordance and relative validity across socioeconomic groups? Perhaps the survey food frequency modules work for some groups and not others. I am most interested in knowing how marital status in Burkina Faso versus Vietnam may have affected recall. I could imagine for married couples that were both participating in the survey in Burkina Faso that their recall may be better if they were interviewed at the same time.

4. The Bland Altman plots suggest there are some outliers in both the Burkino Faso and Vietnam samples. I wonder how much the outliers may be driving some of the analysis and what would happen if you drop them.

5. The author(s) note some differences between the short- and long-list samples in Vietnam, specifically differences in education. In particular, the long-list module appears to have a higher prevalence of adults with more education. Could this have influenced the results at all?

6. It would be helpful if the author(s) included in the appendix the list of foods in long and short list food frequency questionnaires and how they aggregated the long-list foods into their short-list counterparts.

7. Not sure if figure 1 really adds much the manuscript. The author(s) succinctly lay out the administration of the data collection in the following paragraph so figure 1 seems superfluous.

8. The three missing observations in table 1 for the Vietnam was due to incomplete responses over the seven days? Please note the specifics somewhere in the text or the table.

9. It is difficult to compare differences between Vietnam and Burkina Faso in the text around p. 13 because the samples begin at very different bases. I recommend talking about the differences in text in terms of percents and referring to the level estimates in table 3. Otherwise, it appeared the Burkina Faso mean differences in energy and macronutrients were much bigger than those in Vietnam.

10. For table 2, I recommend adding a total concordance row and a total non-concordance row.

11. I very much liked the discussion as it gives some guidance for other researchers in constructing the food frequency modules on household consumption and expenditure surveys. But I do think the discussion could also compare and contrast a bit more heavily from lessons learned from other studies that conducted similar analysis in developing countries, some of which are listed in comment 2.

12. For the Bland Altman plots, I would prefer the mean difference to be labeled directly on the dashed line, probably also the confidence interval values could also appear directly on the plot. It was annoying going between the text above the plot to the plot to place the information.

13. Just a handful of grammar issues that should be addressed:

a. P. 3, line 57: “In high-income countries, it has been shown thant…”

b. P. 4, line 77: “…information collected remains minimumal.”

c. P. 4, line 95: “…we developed over a one-week period…”

d. P. 7, line 162: “…conducted with in the same participants…”

e. P.9, line 228: “Mmonetary expenditure…”

f. P. 18, last paragraph: misspelled completely

References

Coates, Jennifer, Beatrice Lorge Rogers, Alexander Blau, Jacqueline Lauer, and Alemzewed Roba. "Filling a dietary data gap? Validation of the adult male equivalent method of estimating individual nutrient intakes from household-level data in Ethiopia and Bangladesh." Food policy 72 (2017): 27-42.

Karageorgou, Dimitra, Fumiaki Imamura, Jianyi Zhang, Peilin Shi, Dariush Mozaffarian, and Renata Micha. "Assessing dietary intakes from household budget surveys: a national analysis in Bangladesh." PLoS one 13, no. 8 (2018): e0202831.

Sununtnasuk, Celeste, and John L. Fiedler. "Can household-based food consumption surveys be used to make inferences about nutrient intakes and inadequacies? A Bangladesh case study." Food Policy 72 (2017): 121-131.

Tang, Kevin, Katherine P. Adams, Elaine L. Ferguson, Monica Woldt, Jennifer Yourkavitch, Sarah Pedersen, Martin R. Broadley, Omar Dary, E. Louise Ander, and Edward JM Joy. "Systematic review of metrics used to characterise dietary nutrient supply from household consumption and expenditure surveys." Public Health Nutrition 25, no. 5 (2022): 1153-1165.

Reviewer #2: Review of PONE-D-24-21954

Assessing food consumed away from home in low-and middle-income countries by developing specific modules for household surveys: experimental evidence from Vietnam and Burkina Faso

This is a thoroughly described and clearly presented manuscript describing an experimental approach to testing alternative questionnaire modules for assessing food consumed away from home (FCAH) in the context of dietary and consumption/expenditure surveys. The rationale is well explained, the methods adequately described, and the results are clearly presented and link directly back to the research question. Limitations are clearly explained.

In addition, I commend the authors for stating definitively that neither approach (long form nor short form) could be validated against the benchmark, three non-consecutive 24 hour dietary recalls conducted during the week at the end of which the FACH module(s) were administered.

My recommendation is that the manuscript be published.

I would not make publication contingent on any revisions or modifications. That said, the authors could note in the methods section that administering the module at the end of the week in which three 24 HR were administered might have sensitized the respondents to be more aware of their food consumption behaviors during the week – a consideration that would only tend to make the FACH module more consistent with the 24 HR than it might otherwise be. There is no alternative to using this procedure for validation, so it would just be a matter of mentioning this possible bias.

I noted a very few places with very minor grammar or spelling errors:

P 18 sixth line: reproductive age, not reproducible age

P 21 last line: to have eaten, not to have eat

P 21 used ‘tented’ for ‘tended’

Spelling of ‘consummed’ should be ‘consumed’

P 12 l 239 capitalize Spearman

None of these jeopardizes the value of the manuscript.

6. PLOS authors have the option to publish the peer review history of their article (what does this mean?). If published, this will include your full peer review and any attached files.

Reviewer #1: No

Reviewer #2: **Yes: **Beatrice Lorge Rogers

---

## [Author Response · Author response to Decision Letter 0]

5 Nov 2024

Academic editor

This has been checked.

We have downloaded as supporting information the English version of the questionnaires that were used for our study.

We do apologize for the mistake, this has been corrected.

4. In the online submission form, you indicated that "Data described in the manuscript, code book, and analytic code will be made available upon request pending application and approval by the authors of the current study."

The present study involved human beings and the data collected are personal. In the protocol approved by both ethics board, there is nothing stating that the data will be made publicly available. In the case of Vietnam the data are owned by the country and permission from the Ministry of Health is required for free and public access to the data. Furthermore, in the information letter given to each partipants it is stated : « The information you provide will not be divulged: only the people involved in this survey will have access to the information collected, anonymously. »

We made the changes to the manuscript accordingly i.e. we do not refer anymore to data that are not part of the manuscript.

This has been added at the end of the manuscript as requested.

The references have been checked.

Reviewer 1

I generally have a favorable impression of this manuscript. The author(s) do a good job of motivating the need to develop measures of dietary intake from food away from home in low- and middle-income countries using household consumption and expenditure surveys. The design of two survey modules (long- and short-form food frequency questionnaires) to be used to approximate dietary intake in household consumption and expenditure surveys, sample design and recruitment, and administration of both the survey modules and 24-hour recalls are described well and seem mostly appropriate although I have some clarifying questions on their approach described below. Below are more detailed comments for the author(s) to consider.

Answer: We thank the reviewer for this general comment about our work and the pertinent and constructive comments.

1. Its not surprising that the weighted average 24-hour recall estimates of energy and macronutrient intake are almost double that of what is reported in a 7-day recall food frequency questionnaire. The choice of 7-day recall versus 24-hour recall seems a bit of an apples to oranges comparison and I’m glad that the author(s) examine this a bit in the Discussion section. But I wonder why the author(s) chose a 7 day recall rather than a diary approach with their survey modules? Also, it seems like quite a burden to collect both the 24-hour recall and food frequency survey modules in the 3rd visit. I wonder how that affected the responses to the food frequency questionnaire and the recall.

Answer: We have chosen repeated 24-hour dietary recalls rather than a 7-day food diary as a reference method because it has been shown that diary-keeping surveys are burdensome for respondents, potentially leading to bias from diary fatigue, and they are many benefits of recall over diary data collection for food items. Furthermore, this approach would have been very difficult to implement in Burkina Faso, given the high prevalence of illiteracy in our sample. 

We agree with the reviewer that collecting both a 24-hour recall and a food away from home module in the third appointment may have increased the burden for participants, leading to a possible fatigue and to report less their food consumption in order to finish the interview more quickly. On the other hand, the participants in the study gained in ability to remember their food intake and estimate the quantities consumed over the course of the various 24-hour recalls, which implies that the burden could have been reduced. In the end, it is difficult to assess the extent to which the responses have been affected in terms of meaning and magnitude. 

2. The author(s) do a pretty good job of summarizing the literature but it seems like there have been a handful of studies not included that have examined the issue of validating household consumption and expenditure data in measuring dietary intake (e.g., Coates et al. 2017; Sununtnasuk and Fiedler 2017; Karageorgou et al 2019; tang et al. 2022). I encourage the author(s) to look at these articles as it seems like many of the points raised in the discussion overlap with the central points of these articles.

Answer: We thank the reviewer for this comment and for the suggestion of papers related to our study. We have included them in the introduction (line 69) as well as in the discussion

3. The author(s) conclude in the abstract and introduction that “None of the developed food away from home modules were considered valid,” which is a bit too strong. The measures were only validated using static measures, i.e., mean comparisons in one time period. However, I wonder how well the survey modules in measuring changes in energy and nutrient composition of food away from home consumption over time? It seems to me that usually we are interested in capturing changes in energy and nutrient consumption. Also, is heterogeneity in concordance and relative validity across socioeconomic groups? Perhaps the survey food frequency modules work for some groups and not others. I am most interested in knowing how marital status in Burkina Faso versus Vietnam may have affected recall. I could imagine for married couples that were both participating in the survey in Burkina Faso that their recall may be better if they were interviewed at the same time.

Answer: Thanks a lot for these comments. The modules developed incorporated few seasonal foods such as fruit or vegetables and when mentioned in the modules they are broadly grouped. Also, in Burkina Faso when married couples both participated in the study they were interviewed separately. 

We run analyses to look at the validity according to gender, marital status or age (male vs. female, under 35 vs. over 35, and single vs. married) and we conclude that the stratification of analyses does not modify our conclusions for energy intake or only slightly modified them for monetary expenditure. 

We understand that our conclusions seem too strong and we slightly changed it in the manuscript (see lines 372-373). 

4. The Bland Altman plots suggest there are some outliers in both the Burkina Faso and Vietnam samples. I wonder how much the outliers may be driving some of the analysis and what would happen if you drop them.

Answer: We removed the outliers and re run the analyses. When the outliers defined in the Bland-Altman analyses are removed (around 5%), the results remain as expected. Indeed, we still have an underestimation of energy intake and an overestimation of expenditure on the part of the module. We can, however, note that this slightly reduces the magnitude of the difference.

5. The author(s) note some differences between the short- and long-list samples in Vietnam, specifically differences in education. In particular, the long-list module appears to have a higher prevalence of adults with more education. Could this have influenced the results at all?

Answer: In both countries, we have a difference of around 10% in the proportions of participants with a high level of education between the short-list and long-list modules (respectively 70% and 82% for Vietnam and 21% and 11% for Burkina Faso, significant for Vietnam but not for Burkina Faso). We believe that this difference has little influence on our results, given that the difference is only 10%. In addition, in both countries we found that the long-list modules performed slightly better than the short-list modules, even though the proportion participants with a high level of education is higher in the short-list module in Burkina Faso and higher in the long-list module in Vietnam.

6. It would be helpful if the author(s) included in the appendix the list of foods in long and short list food frequency questionnaires and how they aggregated the long-list foods into their short-list counterparts.

Answer: Thanks a lot. The long and short lists for both countries have been added to the supplementary materials.

7. Not sure if figure 1 really adds much the manuscript. The author(s) succinctly lay out the administration of the data collection in the following paragraph so figure 1 seems superfluous.

Answer: We have removed Figure 1.

8. The three missing observations in table 1 for the Vietnam was due to incomplete responses over the seven days? Please note the specifics somewhere in the text or the table.

Answer: Socio demographic characteristics of participants were asked just once during the first visit. For three participants, age was not recorded because either the enumerator forgot to enter the data, or to ask, or the participant did not answer the question, this why the sample size for age is 415 and not 418 as for the rest of the variables.

9. It is difficult to compare differences between Vietnam and Burkina Faso in the text around p. 13 because the samples begin at very different bases. I recommend talking about the differences in text in terms of percents and referring to the level estimates in table 3. Otherwise, it appeared the Burkina Faso mean differences in energy and macronutrients were much bigger than those in Vietnam.

Answer: We thank the reviewer for the great comment. We added in the text the translation of the differences with the 24-hour dietary recalls in percent (see lines 305-307 and 343-344). It shows that the differences are greater for Vietnam than for Burkina Faso for the budget allocated to food consumed away from home, whereas they are similar for the energy intake coming from foods consumed away from home.

10. For table 2, I recommend adding a total concordance row and a total non-concordance row.

Answer: We have updated the table 2.

11. I very much liked the discussion as it gives some guidance for other researchers in constructing the food frequency modules on household consumption and expenditure surveys. But I do think the discussion could also compare and contrast a bit more heavily from lessons learned from other studies that conducted similar analysis in developing countries, some of which are listed in comment 2.

Answer: We thank the reviewer for that comment. We have added 2 sentences in the discussion based on the references suggested in comment 2 (see lines 360-363).

12. For the Bland Altman plots, I would prefer the mean difference to be labelled directly on the dashed line, probably also the confidence interval values could also appear directly on the plot. It was annoying going between the text above the plot to the plot to place the information.

Answer: We have modified the Bland and Altman plots so that now the mean difference as well as the confidence interval values directly appear on the plots.

13. Just a handful of grammar issues that should be addressed

a. P. 3, line 57: “In high-income countries, it has been shown thant…”

b. P. 4, line 77: “…information collected remains minimumal.”

c. P. 4, line 95: “…we developed over a one-week period…”

d. P. 7, line 162: “…conducted with in the same participants…”

e. P.9, line 228: “Mmonetary expenditure…”

f. P. 18, last paragraph: misspelled completely

Answer: We thank the reviewer for highlighting grammatical and spelling errors. We have modified the manuscript to correct them. 

Reviewer 2

Assessing food consumed away from home in low-and middle-income countries by developing specific modules for household surveys: experimental evidence from Vietnam and Burkina Faso

This is a thoroughly described and clearly presented manuscript describing an experimental approach to testing alternative questionnaire modules for assessing food consumed away from home (FCAH) in the context of dietary and consumption/expenditure surveys. The rationale is well explained, the methods adequately described, and the results are clearly presented and link directly back to the research question. Limitations are clearly explained.

In addition, I commend the authors for stating definitively that neither approach (long form nor short form) could be validated against the benchmark, three non-consecutive 24 hour dietary recalls conducted during the week at the end of which the FACH module(s) were administered.

My recommendation is that the manuscript be published.

I would not make publication contingent on any revisio

---

## [Editor Report · Decision Letter 1]

11 Nov 2024

PONE-D-24-21954R1Assessing food consumed away from home in low-and middle-income countries by developing specific modules for household surveys: experimental evidence from Vietnam and Burkina FasoPLOS ONE

Dear Dr. Landais,

Thank you for submitting your manuscript to PLOS ONE. After careful consideration, we feel that it has merit but does not fully meet PLOS ONE’s publication criteria as it currently stands. Therefore, we invite you to submit a revised version of the manuscript that addresses the points raised during the review process. Thank you for your careful response to reviewer comments. I have noted a few small typos and made suggestions for small language edits, and that the paper proceed to publication once these changes are made. You do not need a detailed rebuttal letter - simply stating that you have accepted the editorial suggestions would suffice (or in case you didn't accept some, just state that).

We look forward to receiving your revised manuscript.

Kind regards,

Susan Horton

Academic Editor

PLOS ONE

Journal Requirements:

Additional Editor Comments:

I am happy that the authors have responded to the reviewers’ comments, and I think this will be a useful addition to the literature. I would like to recommend some modifications to the language/typos prior to publication.

Line 25: “has” not “have”

Line 60: “limited existing empirical evidence” not “few existing empirical evidences”

Line 138: “per module” not “per modules”

Line 276 and 305: “protein, carbohydrate and fat intake” not plural form (but note that the plural form is ok in other places e.g. line 332 when “intake” is not included.

Line 370: remove the 2 extra commas

Line 372 “right category of a consumer of food away from home” would be better

Line 375: “omitted” not “omission”

Line 390: “areas” not “area”

Line 411: “there” not “they”

Line 417: I am not sure FCAFH has been defined and should not be introduced here. FAFH is a well-known acronym and I would recommend introducing this on line 49 and using throughout the manuscript, and also use this at some points e.g. line 414 where the phrase has been shortened to “away from home”

Line 424: “their” (typo)

Line 430: “recall” not “recalling”

Line 436: “sizes” not “size”

Line 444: “relative” not “relatively”

---

## [Author Response · Author response to Decision Letter 1]

12 Nov 2024

Academic editor

This has been checked.

We have downloaded as supporting information the English version of the questionnaires that were used for our study.

We do apologize for the mistake, this has been corrected.

4. In the online submission form, you indicated that "Data described in the manuscript, code book, and analytic code will be made available upon request pending application and approval by the authors of the current study."

The present study involved human beings and the data collected are personal. In the protocol approved by both ethics board, there is nothing stating that the data will be made publicly available. In the case of Vietnam the data are owned by the country and permission from the Ministry of Health is required for free and public access to the data. Furthermore, in the information letter given to each partipants it is stated : « The information you provide will not be divulged: only the people involved in this survey will have access to the information collected, anonymously. »

We made the changes to the manuscript accordingly i.e. we do not refer anymore to data that are not part of the manuscript.

This has been added at the end of the manuscript as requested.

The references have been checked.

Additional Editor Comments:

I am happy that the authors have responded to the reviewers’ comments, and I think this will be a useful addition to the literature. I would like to recommend some modifications to the language/typos prior to publication.

Line 25: “has” not “have”

Line 60: “limited existing empirical evidence” not “few existing empirical evidences”

Line 138: “per module” not “per modules”

Line 276 and 305: “protein, carbohydrate and fat intake” not plural form (but note that the plural form is ok in other places e.g. line 332 when “intake” is not included.

Line 370: remove the 2 extra commas

Line 372 “right category of a consumer of food away from home” would be better

Line 375: “omitted” not “omission”

Line 390: “areas” not “area”

Line 411: “there” not “they”

Line 417: I am not sure FCAFH has been defined and should not be introduced here. FAFH is a well-known acronym and I would recommend introducing this on line 49 and using throughout the manuscript, and also use this at some points e.g. line 414 where the phrase has been shortened to “away from home”

Line 424: “their” (typo)

Line 430: “recall” not “recalling”

Line 436: “sizes” not “size”

Line 444: “relative” not “relatively”

We have accepted the editorial suggestions regarding typos and small edits suggestions.

Reviewer 1

I generally have a favorable impression of this manuscript. The author(s) do a good job of motivating the need to develop measures of dietary intake from food away from home in low- and middle-income countries using household consumption and expenditure surveys. The design of two survey modules (long- and short-form food frequency questionnaires) to be used to approximate dietary intake in household consumption and expenditure surveys, sample design and recruitment, and administration of both the survey modules and 24-hour recalls are described well and seem mostly appropriate although I have some clarifying questions on their approach described below. Below are more detailed comments for the author(s) to consider.

Answer: We thank the reviewer for this general comment about our work and the pertinent and constructive comments.

1. Its not surprising that the weighted average 24-hour recall estimates of energy and macronutrient intake are almost double that of what is reported in a 7-day recall food frequency questionnaire. The choice of 7-day recall versus 24-hour recall seems a bit of an apples to oranges comparison and I’m glad that the author(s) examine this a bit in the Discussion section. But I wonder why the author(s) chose a 7 day recall rather than a diary approach with their survey modules? Also, it seems like quite a burden to collect both the 24-hour recall and food frequency survey modules in the 3rd visit. I wonder how that affected the responses to the food frequency questionnaire and the recall.

Answer: We have chosen repeated 24-hour dietary recalls rather than a 7-day food diary as a reference method because it has been shown that diary-keeping surveys are burdensome for respondents, potentially leading to bias from diary fatigue, and they are many benefits of recall over diary data collection for food items. Furthermore, this approach would have been very difficult to implement in Burkina Faso, given the high prevalence of illiteracy in our sample. 

We agree with the reviewer that collecting both a 24-hour recall and a food away from home module in the third appointment may have increased the burden for participants, leading to a possible fatigue and to report less their food consumption in order to finish the interview more quickly. On the other hand, the participants in the study gained in ability to remember their food intake and estimate the quantities consumed over the course of the various 24-hour recalls, which implies that the burden could have been reduced. In the end, it is difficult to assess the extent to which the responses have been affected in terms of meaning and magnitude. 

2. The author(s) do a pretty good job of summarizing the literature but it seems like there have been a handful of studies not included that have examined the issue of validating household consumption and expenditure data in measuring dietary intake (e.g., Coates et al. 2017; Sununtnasuk and Fiedler 2017; Karageorgou et al 2019; tang et al. 2022). I encourage the author(s) to look at these articles as it seems like many of the points raised in the discussion overlap with the central points of these articles.

Answer: We thank the reviewer for this comment and for the suggestion of papers related to our study. We have included them in the introduction (line 69) as well as in the discussion

3. The author(s) conclude in the abstract and introduction that “None of the developed food away from home modules were considered valid,” which is a bit too strong. The measures were only validated using static measures, i.e., mean comparisons in one time period. However, I wonder how well the survey modules in measuring changes in energy and nutrient composition of food away from home consumption over time? It seems to me that usually we are interested in capturing changes in energy and nutrient consumption. Also, is heterogeneity in concordance and relative validity across socioeconomic groups? Perhaps the survey food frequency modules work for some groups and not others. I am most interested in knowing how marital status in Burkina Faso versus Vietnam may have affected recall. I could imagine for married couples that were both participating in the survey in Burkina Faso that their recall may be better if they were interviewed at the same time.

Answer: Thanks a lot for these comments. The modules developed incorporated few seasonal foods such as fruit or vegetables and when mentioned in the modules they are broadly grouped. Also, in Burkina Faso when married couples both participated in the study they were interviewed separately. 

We run analyses to look at the validity according to gender, marital status or age (male vs. female, under 35 vs. over 35, and single vs. married) and we conclude that the stratification of analyses does not modify our conclusions for energy intake or only slightly modified them for monetary expenditure. 

We understand that our conclusions seem too strong and we slightly changed it in the manuscript (see lines 372-373). 

4. The Bland Altman plots suggest there are some outliers in both the Burkina Faso and Vietnam samples. I wonder how much the outliers may be driving some of the analysis and what would happen if you drop them.

Answer: We removed the outliers and re run the analyses. When the outliers defined in the Bland-Altman analyses are removed (around 5%), the results remain as expected. Indeed, we still have an underestimation of energy intake and an overestimation of expenditure on the part of the module. We can, however, note that this slightly reduces the magnitude of the difference.

5. The author(s) note some differences between the short- and long-list samples in Vietnam, specifically differences in education. In particular, the long-list module appears to have a higher prevalence of adults with more education. Could this have influenced the results at all?

Answer: In both countries, we have a difference of around 10% in the proportions of participants with a high level of education between the short-list and long-list modules (respectively 70% and 82% for Vietnam and 21% and 11% for Burkina Faso, significant for Vietnam but not for Burkina Faso). We believe that this difference has little influence on our results, given that the difference is only 10%. In addition, in both countries we found that the long-list modules performed slightly better than the short-list modules, even though the proportion participants with a high level of education is higher in the short-list module in Burkina Faso and higher in the long-list module in Vietnam.

6. It would be helpful if the author(s) included in the appendix the list of foods in long and short list food frequency questionnaires and how they aggregated the long-list foods into their short-list counterparts.

Answer: Thanks a lot. The long and short lists for both countries have been added to the supplementary materials.

7. Not sure if figure 1 really adds much the manuscript. The author(s) succinctly lay out the administration of the data collection in the following paragraph so figure 1 seems superfluous.

Answer: We have removed Figure 1.

8. The three missing observations in table 1 for the Vietnam was due to incomplete responses over the seven days? Please note the specifics somewhere in the text or the table.

Answer: Socio demographic characteristics of participants were asked just once during the first visit. For three participants, age was not recorded because either the enumerator forgot to enter the data, or to ask, or the participant did not answer the question, this why the sample size for age is 415 and not 418 as for the rest of the variables.

9. It is difficult to compare differences between Vietnam and Burkina Faso in the text around p. 13 because the samples begin at very different bases. I recommend talking about the differences in text in terms of percents and referring to the level estimates in table 3. Otherwise, it appeared the Burkina Faso mean differences in energy and macronutrients were much bigger than those in Vietnam.

Answer: We thank the reviewer for the great comment. We added in the text the translation of the differences with the 24-hour dietary recalls in percent (see lines 305-307 and 343-344). It shows that the differences are greater for Vietnam than for Burkina Faso for the budget allocated to food consumed away from home, whereas they are similar for the energy intake coming from foods consumed away from home.

10. For table 2, I recommend adding a total concordance row and a total non-concordance row.

Answer: We have updated the table 2.

11. I very much liked the discussion as it gives some guidance for other researchers in constructing the food frequency modules on household consumption and expenditure surveys. But I do think the discussion could also compare and contrast a bit more heavily from lessons learned from other studies that conducted similar analysis in developing countries, some of which are listed in comment 2.

Answer: We thank the reviewer for that comment. We have added 2 sentences in the discussion based on the references suggested in comment 2 (see lines 360-363).

12. For the Bland Altman plots, I would prefer the mean difference to be labelled directly on the dashed line, probably also the confidence interval values could also appear directly on the plot. It was annoying going between the text above the plot to the plot to place the information.

Answer: We have modified the Bland and Altman plots so that now the mean difference as well as the confidence interval values directly appear on the plots.

13. Just a handful of grammar issues that should be addressed

a. P. 3, line 57: “In high-income countries, it has been shown thant…”

b. P. 4, line 77: “…information collected remains minimumal.”

c. P. 4, line 95: “…we developed over a one-week period…”

d. P. 7, line 

---

## [Editor Report · Decision Letter 2]

18 Nov 2024

Assessing food consumed away from home in low-and middle-income countries by developing specific modules for household surveys: experimental evidence from Vietnam and Burkina Faso

PONE-D-24-21954R2

Dear Dr. Landais,

We’re pleased to inform you that your manuscript has been judged scientifically suitable for publication and will be formally accepted for publication once it meets all outstanding technical requirements.

Kind regards,

Susan Horton

Academic Editor

PLOS ONE

Additional Editor Comments (optional):

Thank you for your patience with the revisions.
---

## [Editor Report · Acceptance letter]

20 Nov 2024

PONE-D-24-21954R2 

PLOS ONE

Dear Dr. Landais, 

I'm pleased to inform you that your manuscript has been deemed suitable for publication in PLOS ONE. Congratulations! Your manuscript is now being handed over to our production team.

Kind regards, 

on behalf of

Dr. Susan Horton 

Academic Editor

PLOS ONE